# Dynamic Periodic Event Graphs for multivariate time series pattern prediction

SoYoung Park, HyeWon Lee and Sungsu Lim

Department of Computer Science and Engineering, Chungnam National University, Daejeon, Republic of South Korea



## ABSTRACT

Understanding and predicting outcomes in complex real-world systems necessitates robust multivariate time series pattern analysis. Advanced techniques, such as dynamic graph neural networks, have shown significant efficacy for these tasks. However, existing approaches often overlook the inherent periodicity in data, leading to reduced pattern or event prediction accuracy, especially in periodic time series. We introduce a new method, called dynamic Periodic Event Graphs (PEGs), to tackle this challenge. The proposed method involves time series decomposition to extract seasonal components that capture periodically recurring patterns within the data. It also uses frequency analysis to extract representative periods from each seasonal component. Additionally, motif patterns, which are recurring sub-sequences in the time series data, are extracted. These motifs are used to define event nodes using the representative periods extracted from the seasonal components. By constructing periodic motif pattern-based dynamic bipartite event graphs, we specifically aim to enhance the performance of link prediction tasks, leveraging periodic characteristics in multivariate time series data. Our method has been rigorously tested on multiple periodic multivariate time series datasets, demonstrating over a 5% improvement in link prediction performance for both transductive and inductive scenarios. This demonstrates a substantial enhancement in predictive accuracy and generalization, providing confidence in the technique's effectiveness. Reproducibility is ensured through publicly available source code, enabling future research and applications.

## INTRODUCTION

Multivariate time series analysis is crucial for capturing real-world complexities through the interactions among multiple variables (*Tsay, 2013*). This approach enhances model accuracy, offers richer information for robust predictions, and provides flexibility for modeling diverse scenarios. Analyzing multivariate time series is essential for predicting future patterns, aiding decision-making, and strategy formulation across various domains. For example, in finance, *Torres & Qiu (2018)*, *Aseeri (2023)* forecast stock market price or volatility, and *Zhao, Xie & West (2016)* optimizes investment portfolios. Similarly, in industry, *Sun et al. (2022)* predicts substation equipment temperature, and *Zhai, Yao & Zhou (2020)* forecasts industrial production levels and optimizes inventory. The inherent complexity of time series data requires a comprehensive understanding of interdependencies and multivariate relationships for accurate modeling. Specifically,

Corresponding author
Sungsu Lim, sungsu@cnu.ac.kr

understanding the nonlinear and temporally shifting relationships between different time series is crucial (*Chiang et al., 2024*). Many forecasting models have been developed to tackle complex temporal patterns and dependencies, but validation of their performance across tasks is still lacking. The absence of comprehensive comparisons across tasks further hinders the assessment and selection of the most effective models (*Yin et al., 2019*).

To address these challenges, graph neural networks (GNNs) have emerged as a powerful approach for forecasting in multivariate time series analysis (*Cao et al., 2020*; *Wu et al., 2020*; *Shao et al., 2022*; *Liu et al., 2022*). These models excel at capturing complex, dynamic interactions among multiple variables, making them particularly effective for modeling relationships inherent in time series data. Building on this, dynamic event-based bipartite graphs (*Wu et al., 2022*) have been proposed as a promising technique to incorporate temporal dependencies, thereby enhancing interpretability and predictive power for pattern forecasting. In such bipartite graph models, time series are represented as one type of node, while patterns or events are represented as another type of node. Edges capture temporal relationships between specific time series at given timestamps, enabling explicit modeling of interdependencies. For instance, Event2Graph (*Wu et al., 2022*) successfully integrates these graph structures to model event-event interactions. However, existing approaches often overlook the inherent periodicity in data, which is critical for improving the performance of link prediction tasks in periodic multivariate time series. The primary objective of this study is to construct dynamic bipartite event graphs by leveraging periodic characteristics in multivariate time series and to validate their effectiveness in enhancing link prediction performance. To bridge this gap, it is essential to model the periodic components of time series explicitly. Leveraging techniques such as Seasonal and Trend decomposition using Loess (STL) (*Cleveland et al., 1990*) and Fourier transformation (*Bracewell, 1989*), periodicity can be identified and incorporated into the graph structure to enhance predictive accuracy. For example, while HVAC systems in multiple households might exhibit regular periodic cycles, the energy usage patterns of household appliances (*e.g.*, refrigerators, washing machines, and dishwashers) often vary due to user behaviors and daily routines (*Cetin, Tabares-Velasco & Novoselac, 2014*). Accurately capturing these nuances through periodicity extraction enables a deeper understanding of individual time series and their unique characteristics, improving model generalization across diverse datasets. In this context, the Periodic Event Graphs (PEGs) advances existing methods by integrating periodic and residual components into the graph representation. This dual-focus approach not only improves the ability to model seasonal trends but also accounts for non-periodic variations, ensuring robustness and versatility across datasets with varying characteristics.

By understanding such nuances and extracting relevant periodicities, we can significantly enhance the accuracy of predicting and modeling pattern event nodes within multivariate time series analysis for graph-based analysis. Extracting useful motifs and other temporal patterns is crucial, as it emphasizes the significance of understanding complex interactions and periodic behaviors in multivariate time series data (*Kim & Lim, 2021*). This article proposes a novel approach that automatically detects periodicity in multivariate time series data to generate the dynamic PEGs, referencing prior research

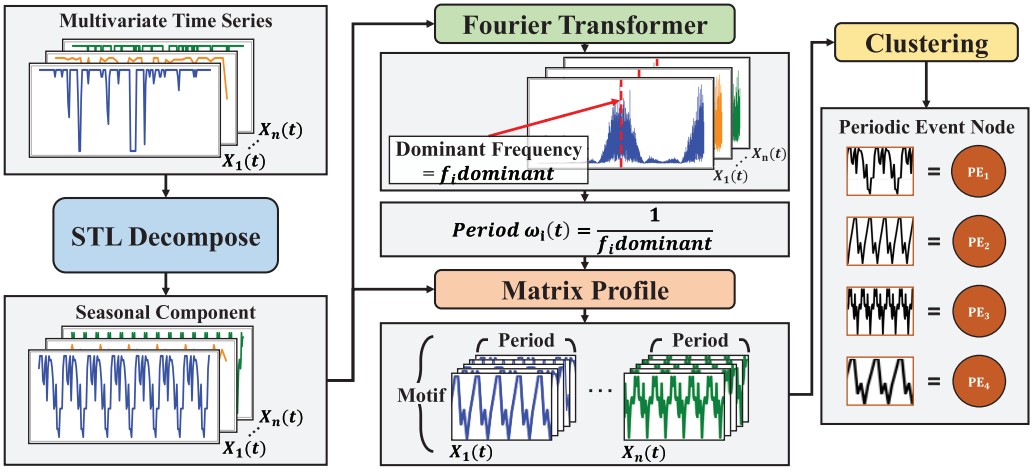

**Figure 1  Multi-step process for generating PEGs.**

(*Wu et al., 2022*). This approach is not just a new method, but a significant advancement in the field of multivariate time series analysis. The proposed approach, illustrated in Fig. 1, generates periodic event nodes based on time series patterns. The approach follows a multi-step process to analyze multivariate time series data. This process involves time series decomposition, frequency analysis, motif extraction, and graph-based representation. First, time series decomposition is applied to each time series to extract seasonal components reflecting inherent periodic patterns. Next, frequency analysis determines the periodicity within each seasonal component, identifying representative periods. These periods are then used to generate representative patterns as motifs, forming the basis for event nodes. Finally, motifs are clustered to create periodic event nodes. This comprehensive process provides insights into the data's periodic nature, enhancing our ability to interpret and understand its underlying patterns.

By extracting seasonal components and identifying periodicity through frequency analysis, we gain a deeper understanding of the temporal dynamics present in the time series data. This understanding is pivotal for developing more accurate predictive models across various domains, allowing us to leverage the periodic patterns for forecasting and decision-making. The potential impact of this research is not limited to one field, but it can revolutionize predictive modeling in many areas. In addition, motifs (*i.e.*, representative patterns) are extracted from each time series using the identified periodicity information. These motifs represent characteristic patterns within the data, serving as the basis for defining pattern-based event nodes. Clustering these motifs across the entire multivariate time series enables the identification of coherent patterns and relationships among different variables. Subsequently, bipartite graphs represent the interactions between these event nodes. This dynamic event graph encapsulates the temporal dependencies and interactions within the multivariate time series data. Such a representation facilitates the effective utilization of various dynamic graph neural networks (DGNNs) for multivariate time series pattern prediction.

The proposed approach offers a systematic and comprehensive framework for analyzing multivariate time series data. It's not just a method, but a complete system for understanding and modeling complex time series data. Integrating time series decomposition, frequency analysis, motif extraction, and graph-based representation, it enables efficient learning of dependencies and accurate modeling of time series for pattern prediction across diverse domains. In summary, our contributions to this work are as follows:

- **Automated time series period extraction:** Automated processes extract periodic patterns from time series data, enhancing model automation and stability.
- **Self-supervised learning for pattern prediction:** Using self-supervised learning, the model autonomously predicts patterns in time series data without manual labeling, enabling efficient use of large datasets. This approach opens up new possibilities for pattern prediction in time series data.
- **Interpretation of patterns as event nodes:** Time series patterns are interpreted as discrete event nodes, improving predictions' understanding, explanation, and reliability.
- **Ensuring reproducibility:** To ensure easy replication of the research, the source code and datasets used in this article are available at https://github.com/peg-repo/periodic-event-graph.

The rest of this article is organized as follows. "Related Work" discusses previous research, "Background and Problem Formulation" provides background and problem definition, "Proposed Method: Periodic Event Graphs" presents our proposed method, "Experiments" describes the experiments and results, and "Conclusions" concludes the article.

## RELATED WORK

Our study focuses on time series pattern or event prediction, aiming to model time series patterns as nodes through GNNs and make predictions *via* link prediction.

Time series pattern prediction plays a pivotal role in uncovering complex temporal behaviors and forecasting future patterns. Traditional methods such as ARIMA, SARIMA, and GARCH have been extensively utilized for analyzing time series data. While these models effectively capture short-term trends and specific periodic features, their ability to manage intricate multivariate interactions or long-term dependencies remains limited. Advances in deep learning have revolutionized this field, enabling the detection and prediction of sophisticated patterns and events. For example, *Zhong, Lv & Shi (2023)* proposed a hybrid model combining long short-term memory (LSTM) networks with conditional random fields (CRFs). This approach excels at capturing long-term dependencies and labeling time series data with structured patterns, although it struggles to explicitly model event-to-event interactions. On the other hand, *Cao et al. (2023)* introduced a generative pre-trained transformer (GPT)-based framework that leverages prompt learning to predict patterns and events. This method demonstrates high efficiency for large-scale data but requires domain-specific fine-tuning.

Recent research has sought to extend time series modeling through graph-based approaches, emphasizing the representation of complex relationships among multivariate data. GNNs have emerged as a prominent tool, enabling explicit relationship modeling and robust pattern learning across datasets. Graph-based methodologies are increasingly applied to analyze and predict complex patterns within multivariate time series data. By representing time series as graph structures—nodes representing entities and edges representing interactions—this approach facilitates an intuitive understanding of relationships. Several studies have advanced the state-of-the-art in this area. *Liu et al. (2022)* developed temporal polynomial graph neural networks (TPGNN), designed to model long-term and intricate dependencies. *Shao et al. (2022)* introduced a pre-training enhanced graph neural network that concurrently learns temporal and spatial interactions, improving prediction accuracy. Similarly, *Wu et al. (2020)* proposed a model integrating graph structures for short- and long-term forecasting. *Cao et al. (2020)* combined frequency-based analysis with graph neural networks in the spectral temporal graph neural network, achieving deep insights into multivariate time series data. DGNNs represent a specialized extension, capturing temporal variations in evolving graphs. Notable DGNN models include JODIE (*Kumar, Zhang & Leskovec, 2019*), which focuses on node interactions and temporal changes; DyRep (*Trivedi et al., 2019*), which learns dynamic embeddings for predictive modeling; and TGAT (*Xu et al., 2020*), which flexibly models temporal graph structures. Additionally, TGN (*Rossi et al., 2020*) and GraphMixer (*Cong et al., 2023*) integrate multi-step graph representations to enhance temporal pattern predictions. The evolutionary state graph by *Hu et al. (2021)*, which models evolving relationships in time series events. This method effectively learns temporal dynamics but overlooks periodic structures within data. *Wu et al. (2022)* addressed this with Event2Graph, an event-driven bipartite graph model integrating temporal dependencies for multivariate time series forecasting and anomaly detection. While Event2Graph excels in modeling event-event interactions, its omission of periodic patterns limits its utility in datasets with pronounced cyclic behaviors. Despite their strengths, these models generally overlook periodic patterns in time series data, leading to suboptimal performance for datasets with strong cyclic characteristics.

Periodic patterns are fundamental to many time series datasets, providing critical insights that enhance predictive accuracy and analytical depth. In this study, we leverage STL and Fourier Transformation to identify and model periodic characteristics in multivariate time series data. This combined approach addresses limitations in traditional methods and offers a robust framework for capturing seasonal and cyclic behaviors. Time series decomposition separates data into trend, seasonality, and residual components, serving as a foundation for various applications. Linear methods such as moving averages and exponential smoothing (*Gardner, 1985*) are intuitive but fail to manage nonlinear complexities. Nonlinear techniques like Wavelet Transformation (*Bentley & McDonnell, 1994*) and singular spectrum analysis (SSA) (*Vautard, Yiou & Ghil, 1992*) have been widely adopted for their ability to extract intricate structures. Among these, STL (*Cleveland et al., 1990*), a non-parametric method based on Locally Estimated Scatter plot Smoothing (LOESS) (*Cleveland, 1979*), excels in isolating localized seasonal patterns, making it

particularly suitable for multivariate time series (*Cao, Qi & Lu, 2024*). Complementing decomposition, frequency analysis identifies recurring patterns through methods like autocorrelation function (ACF), partial autocorrelation function (PACF) (*Box et al., 2015*), cyclical dummy variables (*Hylleberg et al., 1990*), and Fourier Transformation (*Bracewell, 1989*). Fourier Transformation is especially effective for rapid detection of dominant periodic components in the frequency domain. *Chen et al. (2019)* demonstrated its utility in cloud computing environments for detecting high-frequency periodicities. Similarly, *Zhou et al. (2022)* integrated Fourier Transformation with decomposition techniques in their Fedformer model, achieving enhanced temporal pattern understanding and forecasting accuracy.

Addressing this limitation, our proposed PEGs explicitly incorporates periodic components into graph-based structures using time series decomposition methods such as STL (*Cleveland et al., 1990*) and Fourier Transformation (*Bracewell, 1989*). PEGs uniquely models both periodic and residual interactions, enhancing its ability to represent and predict diverse data features. In this study, STL is used to decompose time series data into trend and seasonal components, while Fourier Transformation identifies dominant periodic patterns. These patterns form the basis for periodic event node construction in PEGs, which also integrates residual nodes to capture non-periodic variations. This approach effectively balances periodic and non-periodic data characteristics, offering a comprehensive framework for multivariate time series analysis and prediction.

# BACKGROUND AND PROBLEM FORMULATION

Multivariate time series pattern or event prediction often encounters challenges due to a mixture of periodic and non-periodic patterns. To address this complexity, we propose PEGs, which leverages time series decomposition, frequency analysis, and advanced graph-based techniques to construct dynamic bipartite graphs. This framework aims to capture temporal dynamics more effectively by incorporating both periodic patterns and residual interactions, enabling accurate link predictions in DGNNs. PEGs integrates key methodologies, including time series decomposition to extract trends, seasonality, and residuals, frequency analysis to identify dominant periodic patterns, and Matrix Profile for motif discovery. These methods, combined with the power of DGNNs, provide a robust approach to dynamic multivariate time series modeling.

## Multivariate tiem series

Consider a multivariate time series dataset $X = \{X_1(t), X_2(t), \ldots, X_n(t)\}$, where $X_i(t)$ represents the $i$-th time series at time $t$. Here, $n$ denotes the number of time series, with each series capturing a different feature or measurement over time. The variable $t$ represents a discrete time index, ranging from $t = 1$ to $t = T$, where $T$ denotes the total number of time steps in the dataset. The dataset $X$ forms a matrix of size $T \times n$, where each row corresponds to a specific time step, and each column corresponds to a particular time series. The goal of multivariate time series analysis is to model the temporal dynamics, interactions, and dependencies between these time series to uncover patterns, infer causal relationships, or make accurate predictions.

## Time series decomposition

Time series decomposition separates multivariate time series data into three components: trend, seasonality, and residuals, expressed as:

$$X_i(t) = T_i(t) + S_i(t) + R_i(t), \tag{1}$$

where $X_i(t)$ represents the observed value of the $i$-th time series at time $t$, $T_i(t)$ is the trend component, $S_i(t)$ is the seasonal component, and $R_i(t)$ represents the residuals. This decomposition is essential for isolating and analyzing the underlying patterns, which enhances the accuracy of forecasting and modeling.

A widely used method for time series decomposition is STL (*Cleveland et al., 1990*), which applies LOESS (*Cleveland, 1979*) to iteratively extract the trend $T_i(t)$ and seasonal components $S_i(t)$. The residuals $R_i(t)$ are then computed as:

$$R_i(t) = X_i(t) - (T_i(t) + S_i(t)). \tag{2}$$

STL decomposition is robust against outliers and provides flexibility in capturing a wide variety of time series patterns. This makes it an ideal choice for extracting the seasonal components $S_i(t)$ and residual components $R_i(t)$ from multivariate time series, ensuring effective separation of periodic and non-periodic behaviors.

## Frequency analysis

In this article, frequency analysis is used to identify periodic patterns in the seasonal components of time series data. A widely adopted method for this purpose is Fourier Transform (FT), which converts time series data from the time domain to the frequency domain. For the seasonal components $S_i(t)$ of each time series, the Fourier Transform is applied as follows:

$$S_i(f) = \int_{-\infty}^{\infty} S_i(t) e^{-i2\pi ft} dt. \tag{3}$$

Here, $S_i(t)$ represents the seasonal component of the $i$-th time series at time $t$, where $t$ is the continuous time variable. FT allows for the extraction of frequency components from the time series, enabling the identification of periodic behaviors by analyzing how the time series behaves over time. The dominant frequency $f_{\text{dominant}}$, corresponding to the highest amplitude in the Fourier spectrum, determines the primary period $\omega$ as the inverse of this frequency:

$$\omega_i = \frac{1}{f_i \text{dominant}}. \tag{4}$$

The dominant frequency $f_i$dominant is crucial for identifying the periodic behavior of the seasonal component. Its inverse provides the primary period $\omega_i$, which is used to segment the time series into periodic patterns. These periodic patterns form the basis for constructing periodic event nodes. The combination of FT for period extraction and the decomposition of time series enables the accurate capture of significant periodic features, which are essential for modeling and predicting temporal patterns.

## Matrix Profile

The Matrix Profile (*Yeh et al., 2016*) is a powerful tool used to identify repeating motifs and measure similarities within time series data. It calculates the distance between all subsequences of a time series, highlighting regions that exhibit similar patterns over time. By capturing these repeating subsequences, the Matrix Profile enables the detection of periodic and recurring behavior within the data. This is particularly useful for time series analysis, where such recurring patterns are often the key to understanding temporal dynamics. Formally, the Matrix Profile is defined as the minimum distance between a subsequence starting at position $k$ and all other subsequences starting at position $l \neq k$:

$$Matrix\ Profile(k) = \min_{l \neq k} \text{dist}(S_i(t)[k : k + \omega_i - 1], S_i(t)[l : l + \omega_i - 1]), \tag{5}$$

where $S_i(t)[k : k + \omega_i - 1]$ represents a subsequence of length $\omega_i$ taken from the seasonal component $S_i(t)$ of the time series $X_i(t)$. The value of $\omega_i$ is the dominant period derived from the FT of $S_i(t)$, as defined in Eq. (4). The function $\text{dist}(\cdot, \cdot)$ is a distance measure (typically the Euclidean distance) between two subsequences. The Euclidean distance between two subsequences is computed as:

$$\text{dist}(S_i(t)[k : k + \omega_i - 1], S_i(t)[l : l + \omega_i - 1]) = \sqrt{\sum_{m=0}^{\omega_i - 1} (S_i(t)[k + m] - S_i(t)[l + m])^2}. \tag{6}$$

Here, the distance measures the similarity between two subsequences by calculating the squared differences between corresponding elements. Once motifs (repeated patterns) are discovered using the Matrix Profile, these motifs are grouped into a motif set $M$, which is defined as the collection of all detected motifs $M = \{M_1, M_2, \ldots, M_n\}$. Each $M_i$ represents a motif detected within the seasonal component $S_i(t)$. These motifs are clustered into $c$ groups to form periodic event nodes, denoted as $V_{PE} = \{PE_1, PE_2, \ldots, PE_c\}$, as shown in Fig. 1. The clustering process groups similar motifs together, forming nodes that represent the periodic behaviors of the time series. The resulting periodic event nodes serve as the building blocks for constructing PEGs, which model the temporal dynamics of the system. The Matrix Profile, combined with the clustering algorithm, ensures that the periodic structures in the data are efficiently captured. By identifying and grouping recurring patterns, PEGs can accurately represent the periodic nature of the time series, providing a powerful framework for tasks such as link prediction.

## Dynamic time warping

To represent these series in a graph structure, we divide each time series $X_i(t)$ into sliding time windows of dominant period $\omega_i$, creating a set of time series nodes:

$$TS_i = \{ts_{t_0}, ts_{t_1}, \ldots, ts_{t_k}\}, \quad ts_{t_j} = \{X_i(t_j), X_i(t_j + 1), \ldots, X_i(t_j + \omega_i - 1)\}, \quad t_j = t_0 + j \cdot s, \tag{7}$$

In this approach, $ts_{t_j}$ represents the sliding window starting at time $t_j$, where $t_j = t_0 + j \cdot s$ for $j = 0, 1, \ldots, k$. Here, $s$ is the stride, which defines the step size for moving

the sliding window. Each sliding window has a fixed length $\omega_i$, which captures the segment $[t_j, t_j + \omega_i - 1]$ of the time series $X_i(t)$. The first node, $ts_{t_0}$, corresponds to the segment starting at $t_0$, and the subsequent node, $ts_{t_1}$, corresponds to the segment starting at $t_1 = t_0 + s$, and so on. The total number of sliding windows, $k + 1$, is determined by the equation $k = \lfloor (T - \omega_i)/s \rfloor$, where $T$ is the total length of the time series $X_i(t)$. Each node $ts_{t_j}$ represents a specific segment of length $\omega_i$, and together, these nodes form the set $TS_i = \{ts_{t_0}, ts_{t_1}, \ldots, ts_{t_k}\}$, referred to as the time series node set of $X_i(t)$. When considering all variables in the multivariate time series dataset $X = \{X_1(t), X_2(t), \ldots, X_n(t)\}$, the union of all time series node sets forms the set $V_{TS} = \{TS_1, TS_2, \ldots, TS_n\}$. This represents the complete set of time series nodes across all variables. These sliding windows allow the time series $X_i(t)$ to be mapped into a graph structure, where each node captures the temporal features of its corresponding segment.

Dynamic time warping (DTW) is a commonly used technique for measuring the similarity between two temporal sequences, which may vary in time or speed. In this context, DTW is used to measure the similarity between time series nodes $V_{TS}$ and event nodes $V_{PE}$. The DTW distance is calculated recursively as follows:

$$DTW(TS_i, PE_j) = \text{dist}(TS_i, PE[j]) + \min \begin{cases} DTW(TS_{i-1}, PE_j), \\ DTW(TS_i, PE_{j-1}), \\ DTW(TS_{i-1}, PE_{j-1}). \end{cases} \tag{8}$$

Here, $TS_i$ represents the $i$-th time series node from $V_{TS}$, and $PE_j$ represents the $j$-th event node from $V_{PE}$. The function $\text{dist}(\cdot, \cdot)$ is the distance measure between the nodes $TS_i$ and $PE_j$.

## Dynamic graph neural networks

DGNNs model temporal changes in graph structures by encoding evolving relationships between nodes and edges, making them highly effective for learning and predicting temporal dynamics. A dynamic graph is defined as:

$$G = \{G_1, G_2, \ldots, G_T\}, \tag{9}$$

where $G_t = (V_t, E_t)$ represents the graph at time $t$, with $V_t$ being the node set and $E_t$ the edge set. The graph structure evolves over time, reflecting the temporal nature of the data. In PEGs, DGNNs are utilized to predict the link formation probabilities between time series nodes $V_{TS}$ and periodic event nodes $V_{PE}$. The likelihood of a link $P(V_{TS}, V_{PE})$ is modeled as:

$$P(V_{TS}, V_{PE}) = f(V_{TS}, V_{PE}; \theta), \tag{10}$$

where $f$ is a dynamic graph neural network function parameterized by $\theta$. By leveraging both temporal features and graph topology, DGNNs enable effective link prediction. DGNNs are particularly useful in modeling time-varying relationships, capturing both the temporal dependencies and the complex interactions between nodes in the graph. The main advantage of DGNNs is their ability to model the dynamic and complex relationships between nodes over time. This enables PEGs to make more accurate predictions compared

to traditional methods. For example, DGNNs are effective in link prediction task which are essential for analyzing multivariate time series pattern. By combining the temporal features with the structural information in the graph, DGNNs allow PEGs to provide an efficient and scalable solution for modeling periodic patterns and dependencies in time series data.

## Problem definition: link prediction in dynamic bipartite graphs

The main goal of this study is to model temporal relationships within multivariate time series data. To achieve this, dynamic bipartite graphs representing interactions between time series nodes $V_{TS}$ and periodic event nodes $V_{PE}$ are constructed. Using DGNNs, the model predicts link formation probabilities between these nodes, effectively capturing temporal dependencies in multivariate time series. Link prediction serves as the performance evaluation method in this study, addressing challenges in dynamic graph modeling and improving accuracy. PEGs decomposes each time series $X_i(t)$ into seasonal component $S_i(t)$ and residual component $R_i(t)$ using STL decomposition, isolating significant periodic components to capture temporal dynamics. The Fourier Transform is applied to the seasonal component $S_i(t)$ to extract the dominant period $\omega$, followed by the identification of repeating motifs through the Matrix Profile. These motifs are clustered and form periodic event nodes, which are linked to time series nodes using DTW, constructing the dynamic bipartite graph. DGNNs are then employed to predict link formation probabilities between time series nodes $V_{TS}$ and periodic event nodes $V_{PE}$. This process allows for predicting future relationships within the time series data. By modeling both periodic and residual components separately, PEGs improves predictive accuracy, offering more accurate predictions compared to traditional methods. This dual-focus approach makes PEGs a robust and powerful tool for time series pattern or event prediction.

The computational complexity of PEGs is dominated by the time series decomposition, period extraction, motif discovery, and dynamic graph construction steps, with an overall complexity of $O(\max(n^2, n \log n))$. PEGs is efficient for small- to medium-sized datasets, and for larger datasets, GPU acceleration or parallel processing techniques can be employed to reduce computational costs. Furthermore, the training process of DGNNs in PEGs does not introduce additional overhead compared to standard GNN training, ensuring scalability and computational efficiency. PEGs separates periodic components using STL decomposition, applies FT for period extraction, and captures periodic behaviors. The Matrix Profile is used to detect recurring motifs, which are clustered to form periodic event nodes. Then, DTW links time series nodes and event nodes, constructing the dynamic bipartite graph. This dynamic graph enables link prediction, overcoming the limitations of traditional methods that do not consider periodicity. By separately modeling periodic and residual components, PEGs improves interpretability and predictive accuracy, making it a powerful and effective tool for multivariate time series analysis. Link prediction performance further demonstrates PEGs superior capabilities compared to other methods, delivering strong results in a variety of tasks.

## PROPOSED METHOD: PERIODIC EVENT GRAPHS

In this section, we introduce a new model, the dynamic PEGs, a unique approach for modeling multivariate time series. Our proposed method focuses on leveraging periodic patterns in multivariate time series to construct dynamic bipartite event graphs. These graphs are explicitly designed to enhance link prediction performance by capturing temporal dynamics more accurately. This graph structure represents periodic time series patterns as event nodes, opening new avenues for pattern prediction analysis.

To evaluate the effectiveness of this proposed graph structure, we use typical DGNNs for link prediction tasks in pattern forecasting. Our primary objective is to validate the applicability of our new graph structure and explore its potential utility in dynamic graph analysis. This research offers a promising tool for future research and applications in the field. The construction process of the proposed PEG model is a comprehensive series of stages, as illustrated in Fig. 2. This approach ensures the robustness and thoroughness of our research. The pseudocode for constructing and predicting with the dynamic PEGS is provided in Algorithm 1.

- Step 1: *Periodic event nodes*: In this initial step, we decompose each time series from the multivariate time series data to extract their seasonal components. We extract periods based on dominant frequencies for each time series using FT. We then utilize the Matrix Profile technique for motif extraction to identify pattern motifs within each time series. These pattern motifs serve as representative pattern nodes for the multivariate time series dataset. Subsequently, periodic event nodes are generated through a clustering process.
- Step 2: *Dynamic Periodic Event Graphs (PEGs)*: In the second step, we construct sliding time windows for each time series based on the seasonal components and representative periods extracted in Step 1. Each time window represents a time series node at a specific timestamp. Employing a pattern-matching algorithm, the periodic event nodes generated in Step 1 are matched with time series nodes to create the dynamic PEGS.
- Step 3: *DGNNs for link prediction*: The proposed model is trained using DGNNs in the final step. These trained models are then applied to link prediction tasks to facilitate pattern forecasting in multivariate time series data.

### Periodic event nodes

The generation of periodic event nodes involves the extraction of representative patterns from multivariate time series through a series of structured steps. This process includes decomposition into seasonal and residual components, period extraction *via* frequency analysis, motif discovery using the Matrix Profile technique, and clustering to define the final event nodes.

Time series decomposition into seasonal ($S_i(t)$) and residual ($R_i(t)$) components is performed using STL decomposition, as detailed in "Background and Problem Formulation". This decomposition isolates periodic and residual behaviors, forming the basis for subsequent period extraction and pattern analysis. Periods are extracted from the

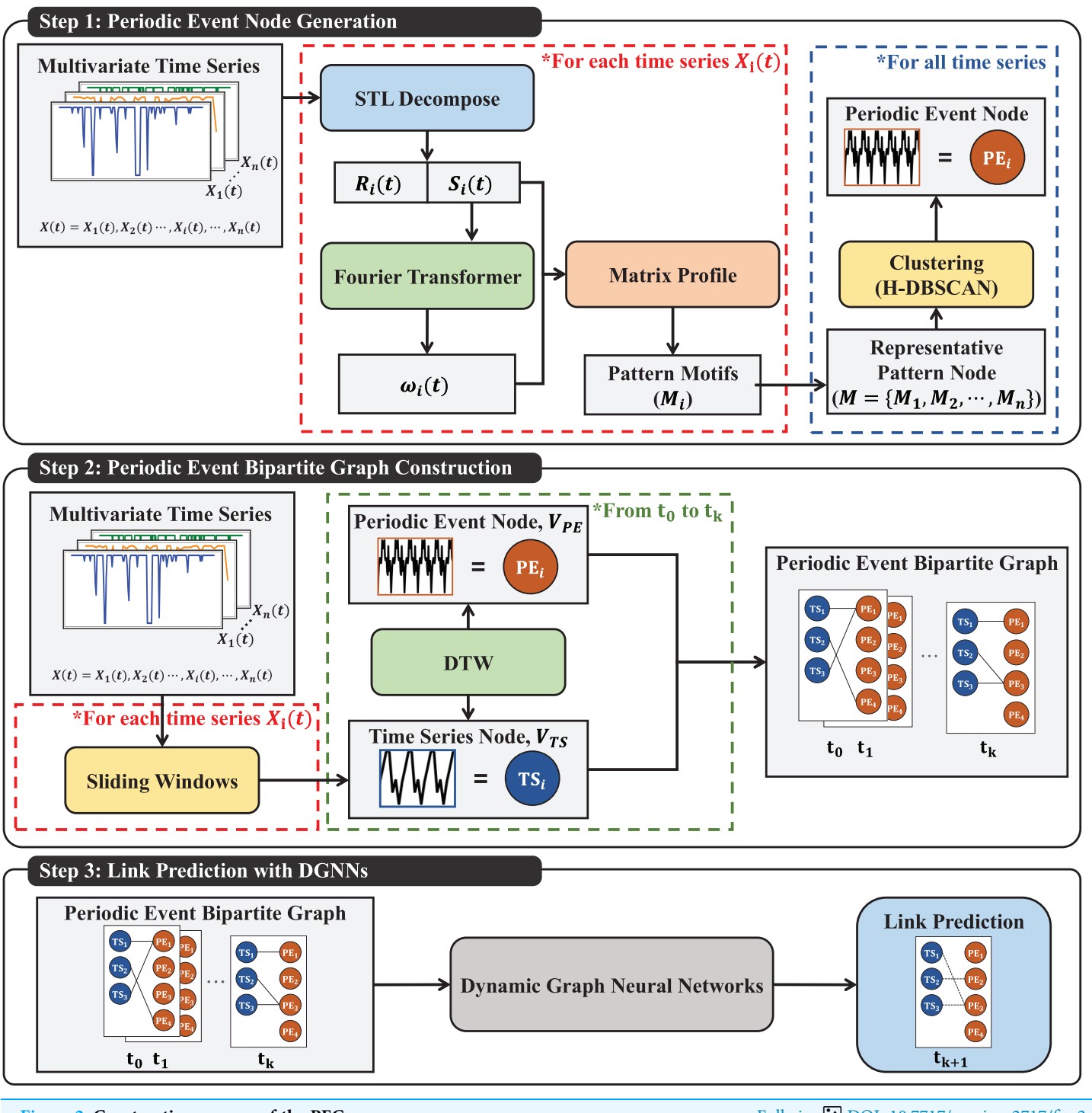

**Figure 2** Construction process of the PEGs.

seasonal components using FT (refer to Eqs. (3) and (4) in "Background and Problem Formulation"). These periods capture dominant temporal characteristics and guide the identification of representative patterns. Motif discovery from seasonal and residual

---

**Algorithm 1** Algorithm for constructing and predicting with PEGs.

**Input:** The algorithm requires a periodic multivariate time series dataset $X = \{X_1(t), \ldots, X_n(t)\}$, where $X_i(t)$ is the $i$-th time series variable, a Dynamic Graph Neural Network (DGNN) model `DGNNs`, and four hyper-parameters: p (STL period), s (stride for sliding window), k (number of motifs), and m (minimum cluster size).

**Output:** Dynamic Periodic Event Graphs $G = \{G_{t_0}, G_{t_1}, \ldots, G_{t_k}, G_{t_{k+1}}\}$, where $G_{t_{k+1}}$ represents the predicted graph containing new or missing links inferred through link prediction.

1: **Step 1: Periodic Event Node Generation**

2: **for** each $X_i(t) \in X$ **do**

3:    $S_i(t), R_i(t) \leftarrow \text{STLDecompose}(X_i(t), \text{p})$             ▷ Decompose $X_i(t)$ using Eq. (1) with given period

4:    $\omega_i \leftarrow \text{FourierTransform}(S_i(t))$             ▷ Extract dominant period $\omega_i$ using Eqs. (3) and (4)

5:    $M_i \leftarrow \text{MatrixProfile}(S_i(t), \text{k})$             ▷ Extract k motifs using Eq. (5)

6: **end for**

7: $M \leftarrow \{M_1, M_2, \ldots, M_n\}$             ▷ Multivariate time series motifs

8: $V_{PE} \leftarrow \text{Clustering}(M, method = \text{H-DBSCAN}, min\_cluster\_size = \text{m}))$      ▷ Cluster motifs to form periodic event nodes

9: $V_{PE} = \{PE_1, PE_2, \ldots, PE_c\}, \; |V_{PE}| = c$             ▷ Periodic event nodes clustered into $c$ groups

10: **Step 2: Periodic Event Bipartite Graph Construction**

11: **for** each $X_i(t) \in X$ **do**

12:    $TS_i \leftarrow \text{SlidingWindows}(X_i(t), \omega_i), \text{s}$             ▷ Create nodes $TS_i$ using Eq. (7)

13:    $TS_i = \{ts_{t_0}, ts_{t_1}, \ldots, ts_{t_k}\}$             ▷ Time series nodes at each time step

14: **end for**

15: $V_{TS} \leftarrow \{TS_1, TS_2, \ldots, TS_n\}$             ▷ Set of all time series nodes

16: **for** each $k$ in $t$ **do**

17:    $e_t \leftarrow \text{DTW}(V_{TS}, V_{PE})$             ▷ Match using DTW as in Eq. (8)

18: **end for**

19: $E \leftarrow \{e_1, e_2, \ldots, e_t\}$             ▷ Edge for graph

20: $G \leftarrow \text{PEGs}(V_{TS}, V_{PE}, E)$             ▷ Construct periodic bipartite graph using Eq. (9)

21: **Step 3: Link Prediction with DGNNs**

22: $\text{Train}(\text{DGNNs}, G)$             ▷ Train DGNN on the graph for link prediction

23: $G_{t_{k+1}} \leftarrow \text{LinkPredict}(\text{DGNNs}, G)$             ▷ Predict links using Eq. (10)

---

components is conducted using the Matrix Profile technique (refer to Eq. (5) in "Background and Problem Formulation"). This approach highlights recurring subsequences in the time series, which are further processed to select the most representative patterns that align with the identified periods. DTW is used to measure the similarity between patterns, leveraging the robust distance matrix generated from Eq. (8) in "Background and Problem Formulation". Similar patterns are grouped using Hierarchical Density-Based Spatial Clustering of Applications with Noise (H-DBSCAN) (*McInnes, Healy & Astels, 2017*). The centroids of these clusters are defined as event nodes, ensuring comprehensive representation of both seasonal and residual patterns.

## Event bipartite graphs

Constructing PEGs involves establishing connections between periodic event nodes and time series nodes. Time series nodes are derived from sliding window datasets containing timestamps for the seasonal and residual components of each time series data. The window size for the sliding windows is determined based on the periods identified during frequency analysis (see "Background and Problem Formulation"), ensuring alignment with the extracted temporal patterns.

For pattern matching between nodes, DTW, as described in "Background and Problem Formulation", is employed to measure the similarity between time series nodes and periodic event nodes. DTW captures non-linear alignments between sequences and provides a robust distance metric, enabling accurate comparisons of patterns even with temporal distortions. The DTW-generated distance matrix is used to identify the most similar event node for each time series node.

After computing similarity, the most similar seasonal or residual event node is connected to each time series node through attribute edges. This process iteratively matches nodes for each timestamp, constructing a bipartite graph structure. The resulting event bipartite graph effectively captures both periodic and residual patterns present in the multivariate time series.

## DGNNs for link prediction

DGNNs are specialized neural networks designed to capture the dynamic behavior of graphs, adapting to changes in graph structure over time. These networks leverage temporal and structural features of dynamic graphs to model evolving node-edge relationships. This capability makes DGNNs effective tools for tasks such as link prediction in PEGs.

The core of DGNNs is a dynamic backbone that integrates two key components: a graph structure encoder and a time encoder. The graph structure encoder generates initial embeddings for nodes and edges based on the topology of the dynamic graph. Meanwhile, the time encoder processes temporal information to capture evolving relationships over time. Together, these encoders provide a robust representation of the graph's dynamic properties.

DGNNs dynamically update node and edge states based on structural and temporal changes, ensuring accurate modeling of evolving relationships. Temporal features are incorporated into the architecture to predict changes in the graph at each time step, facilitating tasks such as link prediction.

In the context of PEGs, DGNNs predict the formation of links between time series nodes ($V_{TS}$) and periodic event nodes ($V_{PE}$). The probability of a link is modeled as $P(v_{TS}, v_{PE}) = f(v_{TS}, v_{PE}; \theta)$, where $f$ represents the DGNN parameterized by $\theta$ (refer to Eq. (10) in "Background and Problem Formulation"). This enables PEGs to identify and forecast relationships between temporal patterns effectively. The workflow of DGNNs for link prediction is depicted in Fig. 3 and comprises three primary steps:

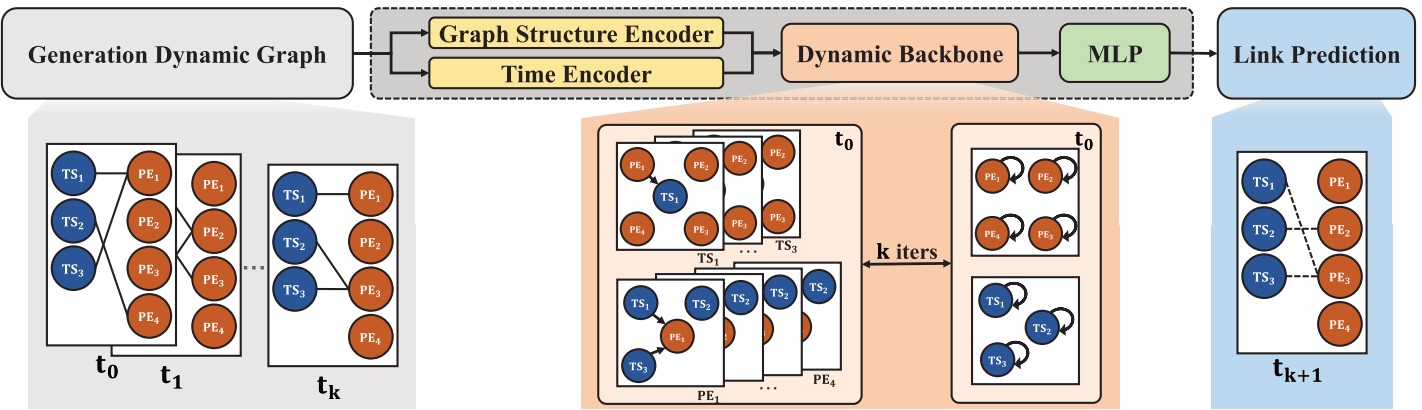

**Figure 3 Workflow of DGNNs for link prediction.**               

1. *Dynamic graph construction:* Using PEGs, the dynamic graph is constructed, with time series nodes (*blue*) and event nodes (*orange*) representing temporal and periodic patterns.

2. *Graph encoding:* The graph structure and temporal features are encoded using the graph structure encoder and time encoder, enabling the dynamic backbone to adaptively model the evolving graph.

3. *Link prediction:* Trained DGNNs are applied to predict links, identifying relationships and patterns in the dynamic graph over time.

### *Evaluation metrics*

Evaluating the performance of link prediction models requires metrics that measure accuracy and discrimination power. Two common evaluation metrics are average precision (AP) and area under the receiver operating characteristic curve (AUC-ROC).

$$AP = \frac{\sum_{k=1}^{n}(p(k) \times rel(k))}{\# \text{ Total positive samples}},$$ (11)

where $P(k)$ represents the precision at cutoff $k$, and $rel(k)$ denotes the relevance of the item at rank $k$. AP quantifies the accuracy of identifying positive samples within the predicted rank list, indicating the model's effectiveness in retrieving positive samples. Higher AP values indicate superior performance.

$$AUC - ROC = \int_{0}^{1} TPR \, (FPR) d \, (FPR)$$ (12)

where $TPR(FPR)$ represents the true positive rate (sensitivity) as a function of the false positive rate (FPR). The AUC-ROC evaluates the model's classification ability, particularly in binary scenarios. A higher AUC-ROC value indicates better model performance, with a value of 1 representing perfect classification. These metrics provide a comprehensive

evaluation of DGNN performance in modeling and predicting dynamic relationships in PEGs, guiding potential research improvements.

# EXPERIMENTS

## Evaluation scenarios

To evaluate the effectiveness of DGNNs with the proposed PEGs for link prediction, we conducted experiments under two distinct scenarios:

- *Transductive*: In the transductive experiments, the model was rigorously trained on a portion of graph data with labeled nodes or edges and made predictions within this dataset to label the remaining unlabeled nodes or edges within the same graph. The evaluation focused on the accuracy of these predictions within the original graph structure, instilling confidence in the thoroughness of our approach.
- *Inductive*: In the inductive experiments, the model was trained on a subset of graph data with labeled nodes or edges and then tested on a separate dataset, potentially containing entirely new examples. The aim was to assess the model's ability to not only learn from the training set but also to generalize its learned patterns to new instances beyond the training set, thereby demonstrating its adaptability and generalization capabilities.

## Experimental settings

### Datasets

*Multivariate time series datasets*

PEGs are constructed using publicly available multivariate time series datasets, including Traffic (*Lai et al., 2018*), Power Consumption (*Goncalves et al., 2022*), and Exchange Rate. The statistics and characteristics of these datasets, including time series length (Length), the number of time series (Number), sampling spacing (Space), and dataset size (Size), mean (Mean), standard deviation (Std Dev.), and temporal patterns (Temporal Pattern), are summarized in Table 1. These datasets represent diverse domains and provide rich temporal features suitable for evaluating PEGs. The datasets used in our experiments span various domains and exhibit diverse temporal characteristics:

- **Traffic** (https://pems.dot.ca.gov): This dataset comprises 48 months of hourly data from 2015 to 2016 collected by the California Department of Transportation. It describes road occupancy rates (between 0 and 1) measured by various freeway sensors in the San Francisco Bay area. The dataset reveals clear weekly peaks and daily diurnal patterns, making it suitable for periodic analysis.
- **Power Consumption** (https://zenodo.org/records/6778401): This dataset contains power consumption data for a local community of 50 households and one public building. The public building data used in the experiment provides consumption profiles segmented by appliances. It spans 96 intervals per day at 15-min intervals, offering a year's worth of data and profiles for 10 appliances. The dataset reflects seasonal and daily variations, capturing appliance-level energy usage trends.

**Table 1 Statistics and characteristics of multivariate time series datasets.**

| Dataset | Length | Number | Space | Size | Mean | Std dev. | Temporal pattern |
|---|---|---|---|---|---|---|---|
| Traffic | 52,560 | 137 | 10 min | 172 MB | 0.35 | 0.12 | Weekly/Hourly Peaks |
| Power consumption | 35,136 | 10 | 15 min | 146 MB | 35.6 kWh | 5.2 kWh | Seasonal (Summer/Winter) |
| Exchange rate | 7,588 | 8 | 1 day | 534 KB | 0.75 | 0.10 | Economic Trends/Events |

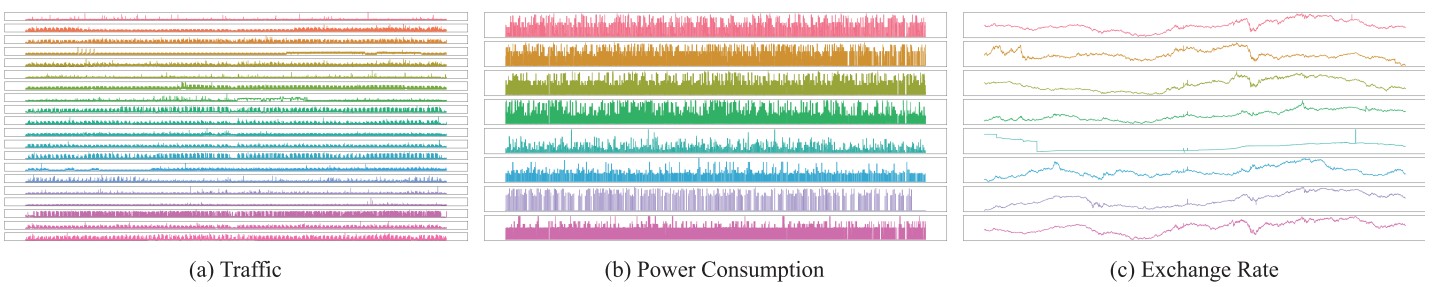

(a) Traffic      (b) Power Consumption      (c) Exchange Rate

**Figure 4 Multivariate time series graphs for three datasets: (A) Traffic dataset, (B) Power Consumption dataset, and (C) Exchange Rate dataset.** Each graph represents the temporal trends and interactions across multiple time series.

- **Exchange Rate:** This dataset includes daily exchange rates from 1990 to 2016 for eight countries: Australia, the UK, Canada, Switzerland, China, Japan, New Zealand, and Singapore. The dataset showcases long-term economic trends and periodic fluctuations influenced by global events and market changes.

The temporal characteristics of each dataset offer a robust foundation for testing PEGs. The Traffic dataset demonstrates distinct diurnal and weekly patterns, Power Consumption captures appliance-level energy trends with seasonal variations, and Exchange Rate reflects long-term economic patterns. These diverse characteristics validate the applicability and effectiveness of PEGs across multiple domains.

Figure 4 illustrates the multivariate time series graphs for three datasets. The Traffic dataset consists of 20 selected time series capturing traffic flow across different locations. The Power Consumption dataset includes 10 time series segmented by appliances, sampled at 15-min intervals to represent detailed energy usage patterns. The Exchange Rate dataset comprises eight time series depicting daily currency exchange rate fluctuations across multiple countries. These datasets were preprocessed using STL decomposition to extract seasonal and residual components, enabling the generation of PEGs with various configurations, as outlined in the methodology.

*Graph datasets generation*
The datasets were first preprocessed to prepare them for analysis. For instance, the Power Consumption dataset was resampled at 15-min intervals, and specific columns were selected for the Traffic dataset to focus on relevant features. STL decomposition (*Cleveland et al., 1990*) was applied to separate the time series data into seasonal and residual

components, isolating periodic patterns for further analysis. Fourier Transform (*Bracewell, 1989*) was then used to analyze the seasonal component, identifying dominant frequencies and extracting representative periods. The Matrix Profile (*Yeh et al., 2016*) was employed to calculate subsequence similarities, enabling the identification of seasonal and residual motifs. These motifs were clustered to generate event nodes, which were subsequently linked to time series nodes representing sliding windows segmented by the extracted periods.

Event graphs (EGs) are constructed without considering periodicity, using a fixed window size to generate event nodes. In contrast, the proposed PEGs incorporates periodicity into the generation of event nodes, resulting in a graph structure that better captures temporal patterns. Additionally, both EGs and PEGs have variations that include residual nodes. Residual nodes are categorized into two types. The first type, simple residual (SR) Nodes, represents only positive or negative deviations from predefined thresholds, capturing unexpected event patterns. The second type, periodic residual (PR) Nodes, is specific to PEGs and is generated by incorporating periodicity into the residual components of time series decomposition. PEGs are designed to capture both periodic and irregular patterns, with variations depending on the inclusion or exclusion of residual nodes. These variations allow for detailed evaluations of the effects of integrating residual components and periodicity into the graph structure. The final PEGs are saved in CSV format for further analysis and experimentation.

For comparative purposes, while Event2Graph (*Wu et al., 2022*) was proposed as a baseline, its implementation code was not publicly available. As a result, we implemented EG independently to replicate a structure that does not account for periodic patterns in the data. These EGs serve as a benchmark to evaluate the advantages of incorporating periodicity in PEGs.

*Graph datasets*

In Table 2, EG is constructed without considering periodicity, using a fixed window size to generate event nodes. In contrast, the proposed PEGs incorporates periodicity to generate event nodes, resulting in a graph structure that better captures temporal patterns. Additionally, there are variations of both EGs and PEGs that include residual nodes. These residual nodes consist of simple residual nodes, which represent only positive or negative deviations, and, in the case of PEGs, periodic residual nodes, which are generated by incorporating periodicity into the residual components of time series decomposition.

Figures 5 and 6 depict the distributions of periodic event nodes and residual nodes within PEGs across timestamps. The histograms provide a visual summary of node occurrences over time:

- Figure 5: Periodic event nodes are distributed based on their periodicity patterns derived from time series decomposition (*e.g.*, seasonal trends).
- Figure 6: Residual nodes capture non-periodic or irregular patterns remaining after periodic components are extracted.

**Table 2 Comparison of nodes and edges between event graph (EG) and Periodic Event Graph (PEG) with residual nodes, classified as simple residual nodes (SR) or periodic residual nodes (PR), leading to various event graph variations.**

| Dataset | Event graph | Residual node | # Nodes | # Edges |
|---|---|---|---|---|
| Traffic | EG | w/o | 24 | 87,680 |
| | | w/SR | 26 | 175,360 |
| | PEG | w/o | 31 | 87,680 |
| | | w/SR | 33 | 175,360 |
| | | w/PR | 40 | 175,360 |
| Power consumption | EG | w/o | 12 | 70,256 |
| | | w/SR | 14 | 140,512 |
| | PEG | w/o | 14 | 70,256 |
| | | w/SR | 16 | 140,512 |
| | | w/PR | 18 | 140,512 |
| Exchange rate | EG | w/o | 13 | 15,160 |
| | | w/SR | 15 | 30,320 |
| | PEG | w/o | 13 | 15,160 |
| | | w/SR | 15 | 30,320 |
| | | w/PR | 16 | 30,320 |

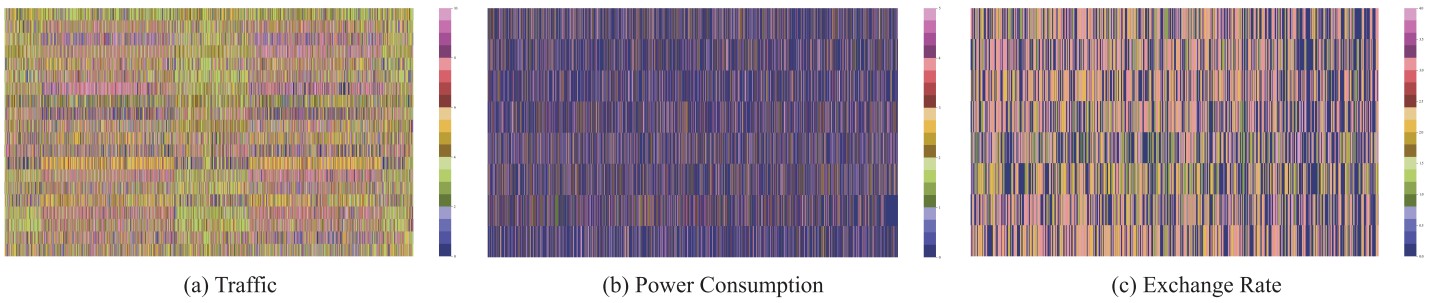

(a) Traffic                    (b) Power Consumption                    (c) Exchange Rate

**Figure 5 Histogram showing the distribution of periodic event nodes in PEGs across timestamps for three datasets: Traffic, Power Consumption, and Exchange Rate.** Colors represent node categories detailed in Table 2.               

The *x*-axis in both figures represents timestamps, and the *y*-axis quantifies the node occurrences at each timestamp. The histogram bars are color-coded to distinguish between event categories, as described in Table 2. For example, in the Traffic dataset, green bars correspond to nodes representing daily traffic patterns, whereas orange bars capture irregular fluctuations, such as anomalies. Similarly, in the Power Consumption dataset, blue bars denote weekly usage patterns, while red bars highlight deviations caused by unusual power demands. By aligning these visualizations with the decomposition and clustering processes of PEGs, Figs. 5 and 6 demonstrate how periodic and residual events are captured and analyzed. These distributions validate the effectiveness of PEGs in distinguishing and modeling different temporal patterns within the datasets.

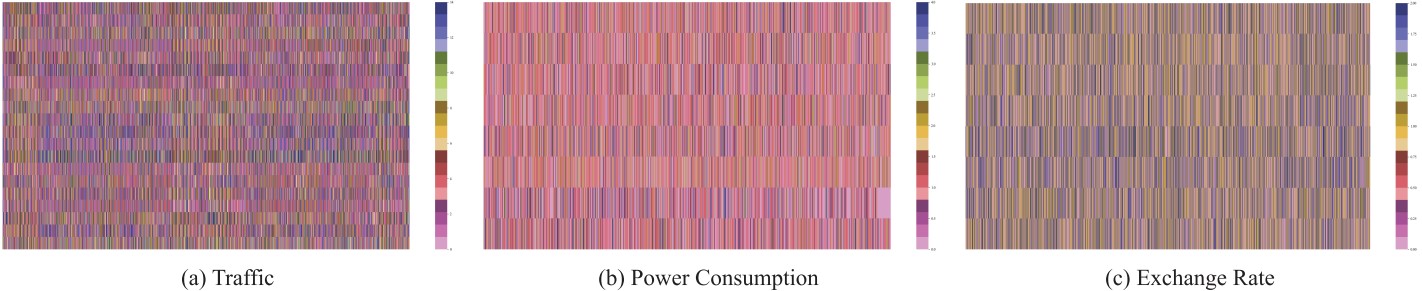

(a) Traffic  (b) Power Consumption  (c) Exchange Rate

**Figure 6 Histogram showing the distribution of residual nodes in PEGs across timestamps for three datasets: (A) Traffic, (B) Power Consumption, and (C) Exchange Rate.** Colors correspond to residual event types defined in Table 2.

## Implementations

In our experimentation, we implemented our methodologies using Python and various external libraries, combining built-in functions with code provided by the authors (*Zhong & Mueen, 2024*; *Siffer et al., 2017*). We utilized the DyGLib (*Yu et al., 2023*) library for dynamic graph neural networks (DGNNs) to conduct comprehensive comparisons between our proposed graphs and existing models, including JODIE (*Kumar, Zhang & Leskovec, 2019*), DyRep (*Trivedi et al., 2019*), TGAT (*Xu et al., 2020*), TGN (*Rossi et al., 2020*), and GraphMixer (*Cong et al., 2023*).

The implementation was carried out in Python 3.9, leveraging several key libraries: PyTorch 1.11.0 for building and training DGNN models, NumPy 1.21.0 and pandas 1.3.3 for numerical operations and data handling, tqdm 4.62.3 for progress visualization, DyGLib for DGNN implementations and link prediction modules, and Matplotlib 3.4.3 for result visualization. All experiments were performed on a server with the following specifications: NVIDIA Tesla V100 GPU (32 GB VRAM), Intel Xeon Gold 6230 CPU (2.10 GHz), 256 GB RAM, 2 TB NVMe SSD, and Ubuntu 20.04 as the operating system.

The standard experimental settings included a batch size of 200, two attention heads, a learning rate of 0.0001, 10 epochs, five runs, a time embedding dimension of 100, a dropout rate of 0.1, and validation and test set ratios of 15%. Negative edge sampling followed a "random" strategy, and historical neighbors were sampled using the "recent" approach to ensure temporal relevance. For each DGNN model, parameters such as the number of neighbors and layers were optimized according to the configurations provided by DyGLib (*Yu et al., 2023*) for link prediction:

- **JODIE:** 10 neighbors, one layer
- **DyRep:** 10 neighbors, one layer
- **TGAT:** 20 neighbors, two layers
- **TGN:** 10 neigTGAT hbors, one layer
- **GraphMixer:** 20 neighbors, two layers

Node features in the event graph were initialized using random initialization techniques, while edge features were generated *via* one-hot encoding to represent the number of nodes connected by each edge. To ensure consistent results and enhance reproducibility, random seeds were fixed across all experiments.

*Reproducibility*

To ensure reproducibility, we have made our code, configurations, and datasets publicly accessible at https://github.com/peg-repo/periodic-event-graph. The repository provides detailed, step-by-step instructions for setting up the environment, executing the experiments, and validating the results, allowing others to replicate our findings with ease.

*Model limitations and constraints*

Our model is specifically designed for multivariate time series data that exhibit periodic patterns. It is particularly effective in tasks such as link prediction, event detection, and temporal relationship analysis, where the identification of periodic events plays a critical role. However, the model has the following limitations and constraints. The model requires input data that include temporal features, node features, and edge features. It performs best on datasets where periodicity is a dominant characteristic. Non-periodic or irregular datasets may result in reduced performance. The scalability of the model depends on the computational resources available. Our experiments were conducted on datasets with up to 33 nodes and 175,360 edges using an NVIDIA Tesla V100 GPU. Larger datasets may require additional memory and computational power. The model is tailored for problems involving temporal dynamics and periodic patterns. For non-temporal graph problems or static data, alternative approaches may be more effective.

### Evaluation metrics

The link prediction performance of dynamic graph neural networks is evaluated using two metrics, AP (Eq. (11)) and AUC-ROC (Eq. (12)), in both transductive and inductive scenarios. Transductive learning performs predictions only within the given graph data, whereas inductive learning generalizes the trained model to make predictions on new data.

## Results

Tables 3 and 4 present the AUC-ROC and AP performance of DGNNs for link prediction tasks in transductive and inductive settings. The evaluation compares various event graph configurations, including the proposed PEGs and standard EGs which differ based on the incorporation of periodicity. EG variations are categorized by residual node types: non-residual (NR), simple residual (SR), and periodic residual (PR) nodes. These experiments were conducted on three datasets "Traffic, Power Consumption, and Exchange Rate" to ensure a comprehensive comparison across diverse domains.

### Transductive experiments

In the transductive DGNNs link prediction experiments, detailed in Table 3, we assessed the performance of different models across three datasets using metrics like AP and AUC-ROC. These models were trained on labeled subsets of nodes or edges and subsequently used to predict labels for the remaining unlabeled nodes or edges within the same graph.

**Table 3 Link prediction performance of transductive DGNNs comparing PEG variants (PEG-NR, PEG-SR, PEG-PR) and EG variants (EG-NR, EG-SR).**

| DGNN | Event graph | Model | Traffic | | Power consumption | | Exchange rate | |
|---|---|---|---|---|---|---|---|---|
| | | | AUC-ROC | AP | AUC-ROC | AP | AUC-ROC | AP |
| JODIE | EGs | EG-NR | 59.87 (0.42) | 59.61 (0.44) | 72.26 (2.21) | 67.53 (1.69) | 50.38 (2.00) | 51.18 (1.75) |
| | | EG-SR | 61.73 (1.69) | 59.50 (1.23) | 64.80 (5.05) | 60.68 (4.55) | 56.77 (1.59) | 55.17 (1.34) |
| | PEGs | PEG-NR | 68.85 (1.82) | 65.62 (1.18) | **75.18 (4.10)** | **71.22 (3.83)** | 62.31 (2.67) | **60.98 (2.31)** |
| | | PEG-SR | 63.86 (1.94) | **79.59 (0.17)** | 70.06 (3.02) | 65.89 (3.38) | 59.21 (2.72) | 59.56 (1.76) |
| | | PEG-PR | **81.89 (0.19)** | 61.17 (1.19) | 71.66 (5.51) | 66.72 (1.89) | **63.62 (2.29)** | 56.64 (1.95) |
| DyRep | EGs | EG-NR | 59.64 (0.34) | 58.06 (0.42) | 72.34 (0.98) | 67.71 (1.50) | 51.45 (2.85) | 51.44 (2.19) |
| | | EG-SR | 62.22 (2.21) | 59.79 (1.45) | 69.48 (1.77) | 66.40 (1.12) | 57.32 (7.98) | 56.24 (6.39) |
| | PEGs | PEG-NR | 66.50 (2.91) | 65.28 (0.98) | **76.14 (1.60)** | **72.64 (0.87)** | 63.54 (1.43) | **62.67 (0.75)** |
| | | PEG-SR | 62.42 (2.46) | **78.17 (1.02)** | 70.61 (2.67) | 59.81 (6.09) | 52.88 (5.41) | 59.43 (3.21) |
| | | PEG-PR | **79.35 (2.02)** | 60.36 (1.86) | 63.54 (8.52) | 67.22 (1.43) | **64.14 (5.47)** | 51.86 (4.34) |
| TGAT | EGs | EG-NR | 79.36 (0.15) | 74.64 (0.31) | 76.15 (0.42) | 72.21 (0.27) | 48.07 (0.58) | 50.05 (0.79) |
| | | EG-SR | 71.62 (1.10) | 66.77 (0.82) | 76.67 (0.45) | 70.53 (0.75) | 50.92 (7.74) | 49.31 (5.25) |
| | PEGs | PEG-NR | 80.38 (0.42) | 78.61 (0.35) | **82.51 (0.05)** | **78.33 (0.10)** | **78.73 (0.22)** | **74.60 (0.28)** |
| | | PEG-SR | 73.81 (0.38) | **82.58 (0.43)** | 76.86 (0.23) | 73.78 (0.22) | 69.87 (0.33) | 70.17 (0.47) |
| | | PEG-PR | **82.72 (0.09)** | 72.17 (0.73) | 80.30 (0.22) | 72.67 (0.24) | 76.26 (0.16) | 66.29 (0.62) |
| TGN | EGs | EG-NR | 75.51 (0.85) | 71.71 (1.08) | 77.14 (0.14) | 72.83 (0.40) | 51.44 (1.67) | 51.65 (0.99) |
| | | EG-SR | 71.43 (0.69) | 66.25 (0.62) | 72.84 (1.11) | 66.29 (0.65) | 53.67 (7.25) | 52.30 (4.96) |
| | PEGs | PEG-NR | 79.06 (0.68) | 77.35 (0.77) | **82.32 (0.09)** | **78.17 (0.14)** | **78.54 (0.77)** | **74.30 (0.80)** |
| | | PEG-SR | 70.24 (1.49) | **81.52 (0.57)** | 75.39 (0.13) | 71.26 (2.39) | 65.47 (1.40) | 69.38 (1.07) |
| | | PEG-PR | **84.53 (0.85)** | 68.60 (1.04) | 78.64 (1.19) | 71.14 (0.35) | 75.94 (0.68) | 60.53 (1.01) |
| GraphMixer | EGs | EG-NR | 77.06 (0.24) | 72.97 (0.34) | 76.15 (0.42) | 72.21 (0.27) | 50.35 (1.01) | 50.36 (1.13) |
| | | EG-SR | 69.29 (2.39) | 65.69 (1.85) | 74.81 (1.28) | 69.58 (0.72) | 50.27 (6.17) | 49.74 (3.92) |
| | PEGs | PEG-NR | 79.49 (0.59) | 77.32 (0.51) | **82.61 (0.17)** | **78.67 (0.13)** | 76.78 (0.80) | **72.37 (0.86)** |
| | | PEG-SR | 72.37 (0.72) | **82.53 (0.30)** | 77.55 (0.46) | 74.07 (0.68) | 69.21 (0.97) | 70.67 (0.41) |
| | | PEG-PR | **86.58 (0.23)** | 71.80 (0.45) | 80.12 (0.39) | 73.28 (0.46) | 76.48 (0.17) | 65.76 (1.33) |

**Note:**
The best result is shown in bold, while the second best is shown in underline.

Across all datasets, PEGs consistently outperformed other representations, with PEG-SR emerging as the top performer. PEG-SR significantly improved over EG-NR and EG-SR in both AP and AUC-ROC scores. Specifically, PEG-SR exhibited an increase of approximately 44% to 53% in AP and 38% to 50% in AUC-ROC compared to EG-NR, and approximately 32% to 33% in AP and 45% to 46% in AUC-ROC compared to EG-SR. This trend persisted across all datasets, with PEG-SR consistently achieving the highest AP and AUC-ROC scores. Notably, PEG-NR showcased the highest scores in the Power Consumption dataset, indicating significant enhancements over EG-NR and EG-SR. Similar results were observed in the Exchange Rate dataset, with PEG-NR demonstrating notable improvements over EGs, particularly in AUC-ROC scores. Furthermore, the transductive experiment showcased our models' adaptability by leveraging intrinsic graph relationships for robust link prediction within known structures. This underscores the

**Table 4  Link prediction performance of inductive DGNNs comparing PEG variants (PEG-NR, PEG-SR, PEG-PR) and EG variants (EG-NR, EG-SR).**

| DGNN | Event graph | Model | Traffic | | Power consumption | | Exchange rate | |
|---|---|---|---|---|---|---|---|---|
| | | | AUC-ROC | AP | AUC-ROC | AP | AUC-ROC | AP |
| JODIE | EGs | EG-NR | 59.47 (4.01) | 54.97 (2.27) | 49.60 (0.33) | 49.94 (0.23) | 51.08 (1.88) | 52.36 (1.70) |
| | | EG-SR | 54.56 (6.60) | 52.03 (4.55) | 49.08 (0.84) | 49.63 (0.64) | **65.53 (4.28)** | **60.10 (4.50)** |
| | PEGs | PEG-NR | **70.73 (2.43)** | 62.38(1.94) | 58.85 (4.81) | 57.64 (3.33) | 50.16 (0.36) | 50.60 (0.55) |
| | | PEG-SR | 56.38 (9.41) | 53.21 (6.28) | 65.29 (3.57) | 60.69 (3.32) | 54.17 (1.79) | 52.97 (1.36) |
| | | PEG-PR | 69.22 (5.05) | **65.34 (4.79)** | **74.10 (2.31)** | **62.67 (1.68)** | 57.30 (2.88) | 55.82 (1.70) |
| DyRep | EGs | EG-NR | 55.47 (7.13) | 51.97 (4.16) | 48.77 (0.70) | 49.39 (0.50) | 50.79 (3.88) | 51.47 (2.93) |
| | | EG-SR | 62.69 (6.39) | 56.53 (5.01) | 47.43 (6.74) | 49.44 (4.78) | 51.82 (19.49) | 53.51 (14.15) |
| | PEGs | PEG-NR | **63.58 (5.51)** | 58.19 (4.08) | **60.93 (14.39)** | **58.13 (8.44)** | 49.84 (0.92) | 50.50 (0.63) |
| | | PEG-SR | 61.58 (6.27) | 55.64 (4.45) | 52.98 (5.19) | 52.87 (2.67) | **57.13 (3.96)** | **54.58 (2.79)** |
| | | PEG-PR | 56.78 (4.46) | **59.48 (4.64)** | 56.38 (18.60) | 54.61 (11.05) | 53.77 (19.53) | 53.79 (11.78) |
| TGAT | EGs | EG-NR | 73.71 (0.22) | 70.00 (0.36) | 49.56 (0.32) | 49.98 (0.22) | 48.23 (1.04) | 50.14 (0.61) |
| | | EG-SR | 63.20 (3.03) | 57.41 (2.12) | 57.90 (1.51) | 54.54 (1.96) | 50.47 (13.27) | 48.29 (8.90) |
| | PEGs | PEG-NR | **80.92 (0.24)** | **77.44 (0.42)** | 73.46 (0.24) | 66.61 (0.43) | 58.70 (0.11) | 55.95 (0.28) |
| | | PEG-SR | 65.66 (0.56) | 61.12 (0.34) | 73.63 (2.46) | 65.38 (2.24) | 65.39 (0.75) | 62.79 (0.87) |
| | | PEG-PR | 80.58 (0.35) | 75.24 (0.42) | **81.56 (0.52)** | **69.75 (1.32)** | **71.03 (0.43)** | **64.35 (0.48)** |
| TGN | EGs | EG-NR | 64.30 (2.09) | 63.14 (2.32) | 49.72 (0.29) | 49.96 (0.22) | 49.53 (2.62 | 50.85 (1.40) |
| | | EG-SR | 64.86 (3.77) | 57.63 (3.43) | 57.55 (0.97) | 52.93 (1.31) | 52.63 (10.00) | 51.13 (7.63) |
| | PEGs | PEG-NR | **79.62 (0.57)** | **76.14 (0.78)** | 72.83 (0.50) | 66.40 (0.18) | 58.85 (0.44) | 56.25 (0.26) |
| | | PEG-SR | 70.20 (4.36) | 64.10 (4.89) | 74.43 (2.31) | 64.43 (2.05) | 64.79 (1.04) | 62.41 (1.31) |
| | | PEG-PR | 78.39 (1.07) | 73.48 (1.29) | **78.91 (1.24)** | **66.69 (2.27)** | **70.53 (0.46)** | **63.65 (0.77)** |
| GraphMixer | EGs | EG-NR | 69.67 (0.89) | 67.58 (0.74) | 49.56 (0.32) | 49.98 (0.22) | 50.75 (2.97) | 51.96 (1.81) |
| | | EG-SR | 58.68 (4.00) | 53.71 (2.59) | 52.76 (1.26) | 49.85 (0.85) | 47.77 (12.06 | 48.15 (7.72) |
| | PEGs | PEG-NR | 78.31 (1.05) | **73.89 (1.20)** | 72.04 (0.31) | 65.62 (0.39) | 58.69 (0.93) | 56.39 (1.00) |
| | | PEG-SR | 60.42 (2.81) | 58.08 (1.46) | 77.53 (1.24) | 68.36 (1.29) | 63.17 (1.36) | 59.87 (1.74) |
| | | PEG-PR | **80.14 (0.59)** | 74.37 (0.64) | **86.36 (2.40)** | **79.40 (4.17)** | **71.38 (0.44)** | **64.66 (0.64)** |

**Note:**
    The best result is shown in bold, while the second best is shown in underline.

effectiveness of integrating PEGs and simple residual nodes in dynamic graph neural networks, validating our approach and suggesting promising directions for future research in dynamic graph modeling. In the transductive setting, paired t-tests revealed that PEG-NR significantly outperformed EGs across all datasets ($p < 0.01$), with PEG-SR also showing improvements over EG-NR ($p < 0.01$) but being less consistent compared to EG-SR ($p < 0.05$); among PEG variants, PEG-PR achieved the best performance, significantly surpassing EGs ($p < 0.01$).

### Inductive experiments

In the inductive experiments, our model was trained on labeled graph data and then tested on a separate dataset, potentially containing new or unseen data, to assess its generalization performance. Summary results are presented in Table 4.

Across various datasets, models with PEGs and simple residual nodes, especially PEG-SR, consistently outperformed others. Notably, in the Traffic dataset, PEG-SR showed significant improvements of around 19% in AP and 16% in AUC-ROC compared to EG-NR, and approximately 25% in AP, and 27% in AUC-ROC compared to EG-SR. Similarly, in the Power Consumption dataset, PEG-SR exhibited remarkable enhancements of approximately 25% in AP and 49% in AUC-ROC over EG-NR, and about 26% in AP and 51% in AUC-ROC over EG-SR. In the Exchange Rate dataset, PEG-SR also performed well, with notable increases of around 6% in AP and 12% in AUC-ROC over EG-NR, and despite a slight decrease in AP, still delivering significant improvement in AUC-ROC compared to EG-SR. These findings underscore the effectiveness of integrating PEGs and simple residual nodes to enhance link prediction performance in dynamic graph neural networks. Moreover, our inductive experiments revealed additional advantages of the model, highlighting its ability to generalize well to new or unseen graph structures, thereby enhancing its applicability in real-world scenarios. Statistical validation for inductive experiments showed that PEG-NR consistently outperformed EGs across all datasets ($p < 0.01$). Paired t-tests further revealed that PEG-SR demonstrated significant improvements over EG-NR ($p < 0.01$) but produced mixed results compared to EG-SR ($p < 0.05$). Among the PEG variants, PEG-PR achieved the most significant advantages over EG-NR and EG-SR ($p < 0.01$), highlighting its effectiveness in inductive scenarios.

## CONCLUSIONS

In this article, we present a novel method called the dynamic PEGs, designed to enhance prediction accuracy in multivariate time series patterns. The proposed method not only enhances model automation and stability but also ensures efficiency by automating the extraction of periodic patterns from time series data. Through self-supervised learning, our model independently identifies patterns in each dataset, eliminating the need for manual labeling and enabling efficient use of large-scale datasets. Representing time series patterns as discrete event nodes enhances interpretability and prediction reliability, facilitating model refinement and decision-making. Experimental results, supported by rigorous statistical validation in both transductive and inductive scenarios, demonstrate that integrating periodicity into event graphs significantly improves link prediction performance for multivariate time series pattern forecasting across diverse domains, including transportation (*e.g.*, predicting traffic flow), power consumption (*e.g.*, forecasting energy demand), and exchange rates (*e.g.*, predicting currency fluctuations). This study's results were rigorously validated through statistical tests, including paired t-tests, demonstrating consistent performance improvements ($p < 0.01$) across multiple scenarios. While PEGs consistently outperforms EGs, its performance may be suboptimal in datasets with minimal or irregular periodicity, where the advantage of periodicity-based modeling diminishes. Further error analysis is necessary to explore such scenarios and optimize performance under these conditions. PEGs exhibit exceptional predictive accuracy, particularly in scenarios without residual nodes. PEG-SR, which introduce simple residual nodes to PEG, effectively capture patterns unexplained by the event graph, further enhancing prediction accuracy. In conclusion, we underscore the transformative

potential of our method in enhancing pattern prediction performance in event graphs based on time series patterns. Future research could focus on applying PEGs to a broader range of multivariate time series datasets, including those with irregular sampling intervals, and exploring its applicability in healthcare, financial forecasting, and environmental monitoring.

### Funding
This work was supported by the National Research Foundation of Korea (NRF) grant funded by the Korea government (MSIT) (No. RS-2023-00214065) and by the Institute of Information & Communications Technology Planning & Evaluation (IITP) grant funded by the Korea government (MSIT) (No. RS-2022-00155857, Artificial Intelligence Convergence Innovation Human Resources Development (Chungnam National University)). This work was also supported by the research fund of Chungnam National University. The funders had no role in study design, data collection and analysis, decision to publish, or preparation of the manuscript.

### Grant Disclosures
The following grant information was disclosed by the authors:
National Research Foundation of Korea (NRF).
Korea government (MSIT): RS-2023-00214065, RS-2022-00155857.
Institute of Information & Communications Technology Planning & Evaluation (IITP).
Artificial Intelligence Convergence Innovation Human Resources Development (Chungnam National University).

### Competing Interests
The authors declare that they have no competing interests.

### Author Contributions
- SoYoung Park conceived and designed the experiments, performed the experiments, analyzed the data, performed the computation work, prepared figures and/or tables, authored or reviewed drafts of the article, and approved the final draft.
- HyeWon Lee analyzed the data, prepared figures and/or tables, authored or reviewed drafts of the article, and approved the final draft.
- Sungsu Lim conceived and designed the experiments, analyzed the data, authored or reviewed drafts of the article, and approved the final draft.

### Data Availability
The code for this study is available at GitHub
- https://github.com/peg-repo/periodic-event-graph.
The data for the experiments is available at GitHub (*Lai et al., 2018*)
- https://github.com/laiguokun/multivariate-time-series-data.

The traffic raw data is available at Caltrans Performance Measurement System (PeMS): https://pems.dot.ca.gov.

The power consumption raw data is available at Zenodo: Goncalves, C., Barreto, R., Faria, P., Gomes, L., & Vale, Z. (2024). Dataset of an Energy Community's Consumption and Generation with Appliance Allocation for One Year [Data set]. Zenodo. https://doi.org/10.5281/zenodo.10854881.

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
