# Peer review of "Dynamic Periodic Event Graphs for multivariate time series pattern prediction"

_PeerJ Computer Science, doi:10.7717/peerj-cs.2717_

## Round 0.1 · original submission · Major Revisions

Please pay particular attention to the concerns of Reviewer 2. As per their comments, the presentation of the article needs to be significantly improved.

Reviewer 1 ·

Basic reporting

- The paper's contribution to the field is reasonable. However, the authors should state clearly and in detail, in the model validation stage, the limitations of model usage (including: type of data, size of data, types of problems where the model is efficient...etc.).
- The authors should refer in their manuscript to their research resources (software, hardware, ... etc.).

Experimental design

No comment.

Validity of the findings

The authors should state clearly and in detail, in the model validation stage, the constraints of their model usage (including: type of data, size of data, types of problems where the model is efficient...etc.).

Additional comments

No comment.

Cite this review as

·

Basic reporting

The manuscript is generally written in professional English, but some parts are ambiguous and require clarification. The introduction mentions the potential contributions of the new Periodic Event Graphs (PEGs) method, but it does not clearly define the scope of the work. Additionally, the abstract and introduction contain keywords suggesting applications like time series forecasting and pattern prediction, but there is no experimental evaluation to support these claims. This results in an unclear context, leaving readers uncertain about the paper's focus until the experimental analysis section.
Similarly to prior work (Wu et al., 2022), the authors should conduct experiments in endpoint analytics such as forecasting or anomaly detection. They need to answer questions like: how much does this new representation improve the current state-of-the-art in these analytics? Without this analysis and experimental evaluation, the paper fails to meet the promises made in the introduction and abstract.
The literature review is inadequate in several respects. The paper references some key studies, but it lacks a detailed comparison with existing methods. For instance, although related work (e.g., Wu et al., 2022) is cited, the manuscript does not properly credit these works nor explain how PEGs differ from them. Furthermore, sections labeled as "related work" mainly describe techniques used in the proposed method rather than positioning PEGs in the current state-of-the-art graph representation methods for time series.
The figures in the manuscript are generally of high quality, but Figures 5 and 6 are difficult to interpret due to a lack of explanation. The purpose of these figures is unclear, as is how the different histograms might have influenced the results. The colors used in the figures are not adequately explained, making interpretation challenging.

Experimental design

The manuscript describes original primary research that seems to fit within the journal’s scope. However, the research question is vaguely defined. While the paper proposes PEGs and their application in link prediction using Graph Neural Networks (GCNs), it does not clearly explain how this research fills a specific knowledge gap in multivariate time series analysis.
Section 4 needs significant improvement. The methodology is not described with sufficient detail to enable replication. The authors should use clear mathematical notation to outline their approach. Concepts such as the use of Matrix Profile, clustering algorithms, and DTW need a more precise description. Additionally, the problem formulation section fails to explain the PEG's objective and how the new proposed method is expected to achieve it.
Furthermore, the manuscript lacks a dedicated background section to explain relevant concepts, resulting in Section 4 containing scattered information about both the methodology and related concepts. This diminishes the clarity and focus on the proposed PEG method.

Validity of the findings

The manuscript's validity is undermined by several issues in the experimental evaluation. The data generation process is not sufficiently described, and the origin and size of the graphs in Table 1 are unclear. It is also not explained how the graphs were constructed, which makes it difficult to assess the robustness of the results. The introduction of the Residual-based EG (REG) model in the experimental section is abrupt and confusing since it lacks a prior explanation of its role and inner workings.
The baseline models used for comparison are not adequately explained. It is unclear whether these are variations of the proposed PEG method or derived from related works. The manuscript also fails to identify which PEG-based model is being proposed by the authors.
The discussion of the results in Tables 2 and 3 includes some inaccuracies. For example, the authors incorrectly claim that "SPEG showcased the highest scores in the Power Consumption dataset" whereas PEG is the best-performing method. This misinterpretation raises concerns about the validity and correctness of the experimental results.
In conclusion, the manuscript's findings are not well-supported due to the lack of transparency in the data generation, inadequate explanations of baseline models, and misinterpretation of results. The conclusions need to be revised to reflect the actual outcomes of the experiments and to be more directly linked to the original research question.

Additional comments

We suggest the authors revise the manuscript to better align with its stated objectives. The introduction and abstract should clearly define the scope and be supported by additional experiments in endpoint analytics, like forecasting or anomaly detection, to validate the proposed method. The related work section should position PEG within existing research more effectively, and the problem formulation and methodology need clearer, more detailed explanations. Use consistent mathematical notation to reduce wordy statements and improve clarity. Additionally, remove redundancy and avoid complex language throughout the paper.

Reviewer 3 ·

Basic reporting

As a reviewer, I suggest several improvements to enhance the paper's quality:

There are several awkward or unclear sentences that should be revised, for example:
Lines 37-40: "Despite numerous forecasting models developed to address these intricate temporal patterns and dependencies, their performance on specific tasks still needs to be more adequately understood." This sentence is convoluted and unclear.
Lines 169-171: The explanation of ωi and its relationship to PEi needs to be clarified for better readability.

The literature review could be strengthened by including more recent work (2023-2024) on decomposition-based methods[1] and dynamic graph neural networks, providing more critical analysis of existing methods' limitations, and adding quantitative comparisons between approaches.

[1] Zhou, Tian, et al. "Fedformer: Frequency enhanced decomposed transformer for long-term series forecasting." International conference on machine learning. PMLR, 2022.

[2] Cao, Defu, et al. "TEMPO: Prompt-based Generative Pre-trained Transformer for Time Series Forecasting." The Twelfth International Conference on Learning Representations.


The mathematical formulation needs more rigor, particularly in explaining the Fourier transform implementation in Equation (2) and elaborating on the probability calculation parameters in Equation (3).

Finally, the methodology section would be strengthened by adding pseudocode for key algorithms, complexity analysis of the proposed method, and formal proofs or theoretical guarantees where applicable.

Experimental design

While the paper presents an original investigation into dynamic periodic event graphs for multivariate time series prediction that aligns with the journal's scope, several key improvements are needed. The research question and knowledge gap regarding the overlooked inherent periodicity in data should be more explicitly articulated with quantitative evidence. Critical methodological details necessary for replication are missing, including specific parameters for time series decomposition, frequency analysis thresholds, motif extraction steps, and clustering implementation details. The claimed 5% performance improvement needs stronger statistical validation through confidence intervals and significance tests. Though code is publicly available, the methodology section should be self-contained with pseudocode and parameter settings. The paper should also address potential limitations and biases in the datasets used for evaluation. These enhancements would significantly strengthen the paper's scientific contribution and reproducibility.

Validity of the findings

The paper's impact and novelty require clearer articulation. While the proposed dynamic Periodic Event Graphs method shows promise with a reported 5% improvement in prediction performance, the paper needs to better demonstrate its practical significance and advantages over existing approaches. The replication study component could be strengthened by explicitly stating how it advances the current literature on multivariate time series analysis. Regarding data robustness, though the authors provided public access to their code, the underlying datasets need more thorough statistical validation - the paper should include detailed statistical tests, error analyses, and control experiments to validate the reliability of results. While the conclusions logically follow from the research findings, they should be more tightly connected to the original research questions about periodic pattern prediction. Additionally, the limitations section should be expanded to acknowledge potential constraints of the proposed method and identify specific scenarios where it might not perform optimally. The paper would benefit from clearer statements about the method's generalizability across different types of time series data and explicit discussion of potential application domains beyond the tested datasets.

Cite this review as

---

## Round 0.2 · accepted · Accept

Thank you for choosing PeerJ Computer Science.

Reviewer 3 ·

Basic reporting

Thanks for the revised version. Currently, all my concerns are addressed.

Experimental design

Thanks for your Table 4 and new added experimental design, which is much clearer than before.

Validity of the findings

Response to Comment 3.6 fully convinced me.

Additional comments

Thanks for the efforts on the revised version, which has above the publish line on my end.

Cite this review as